# High Resistant Starch Rice: Variation in Starch Related SNPs, and Functional, and Sensory Properties

**DOI:** 10.3390/foods11010094

**Published:** 2021-12-30

**Authors:** Ming-Hsuan Chen, Karen Bett-Garber, Jeanne Lea, Anna McClung, Christine Bergman

**Affiliations:** 1Dale Bumpers National Rice Research Center, USDA-ARS, Stuttgart, AR 72160, USA; Anna.Mcclung@USDA.GOV; 2Southern Regional Research Center, USDA-ARS, New Orleans, LA 70124, USA; Karen.Bett@USDA.GOV (K.B.-G.); Jeanne.Lea@USDA.GOV (J.L.); 3Food and Beverage Department, University of Nevada Las Vegas, Las Vegas, NV 89154, USA; Christine.Bergman@UNLV.EDU

**Keywords:** rice, resistant starch, sensory, functional property, RVA, *Waxy*, *SSIIa*

## Abstract

Human diets containing greater resistant starch (RS) are associated with superior glycemic control. Although high amylose rice has higher RS (29 g/kg to 44 g/kg) than lower amylose content varieties, sensory and processing properties associated with RS have not been evaluated. This study used variants of *Waxy* and *starch synthase II a* (*SSIIa*) genes to divide high amylose (256 g/kg to 284 g/kg) varieties into three haplotypes to examine their effects on RS, RVA parameters, and 14 cooked rice texture properties. RVA characteristics were influenced by both genes with peak and hotpaste viscosity differentiating the three haplotypes. Setback from hotpaste viscosity was the only RVA parameter correlated with RS content across three haplotypes (*r* = −0.76 to −0.93). Cooked rice texture attributes were impacted more by *Waxy* than by *SSIIa* with initial starch coating, roughness, and intact particles differentiating the three haplotypes. Pairwise correlation (*r* = 0.46) and PCA analyses suggested that roughness was the only texture attribute associated with RS content; while protein content influenced roughness (*r* = 0.49) and stickiness between grains (*r* = 0.45). In conclusion, variation exists among genetic haplotypes with high RS for sensory traits that will appeal to diverse consumers across the globe with limited concern for negatively affecting grain processing quality.

## 1. Introduction

Dietary fiber consumption is recommended to decrease the risk of developing chronic diseases such as obesity, diabetes, cardiovascular disease, and some cancers [1]. Cereal grains contain diverse types of dietary fiber which have different chemical structures that affect fermentability in the gut, composition of gut microbiota, and the production of metabolites; all of which are thought to positively influence human health [2].

Resistant starch (RS) is a type of dietary fiber in cereal grains that resists digestion in the small intestine and has higher fermentability in the gut than other dietary fibers, such as the non-starch polysaccharides from cell wall material [3]. RS supplements reportedly improved fasting glucose and insulin, insulin sensitivity, and LDL-cholesterol concentration, especially for those with diabetes and obesity [4]. In animal and human models, additional health impacts of RS included ameliorating pancreatic dysfunction, reducing body fat, and improving colon health [5,6].

There are five types of RS (RS I-V). RS I, physically embedded, i.e., in a protein matrix or thick cell-wall; RS II, raw granules or un-gelatinized starch; RS III, retrograded starch; RS IV, chemically or enzymatically modified starch; and RS V, lipid-starch complexes [7,8]. Different host bacteria in the gut at the phyla and species levels favor different RS types and the sub-structures within these [2]. Since rice is consumed after cooking, RS III is the primary form found in cooked rice. The RS III increased the relative abundance of *Faecalibacterium prausnitzii* and reduced the number of *Escherichia coli* and *Pseudomonas* spp. in pigs [9]. *F. prausnitzii* produces butyrate, a short chain fatty acid, which is the major energy source for colonocytes, modulates immune and inflammatory responses and enhances the barrier function of the intestinal epithelium [5,10].

The crystalline structures of RS III are primarily formed from retrograded amylose; therefore, RS in cooked rice is positively correlated with apparent amylose content (AAC) of rice [8,11,12,13]. Recent studies demonstrated the negative correlation of AAC and RS with predicted glycemic index (GI) [12,14,15]. Thus, consumption of high amylose and high RS rice varieties, relative to those that have intermediate or low amylose, are suggested to better manage and reduce the risk of developing type II diabetes [16].

Amylose content, which is positively associated with the firmness of cooked rice, is one of the major quality traits that determines consumers’ preference globally. High amylose varieties are popular in some regions, i.e., South and Southeast Asia, West Africa, and South America [17,18]. However, a majority of rice consumers prefer rice with intermediate and low amylose content [17,18]. However, high amylose varieties widely range in cooked texture and processing properties [19,20,21]. The firmness of some high amylose varieties appears to overlap with those of the other amylose classes suggesting the potential to expand consumption of high amylose varieties to consumers that prefer less firm cooked rice texture [20]. In addition, a particular type of high amylose rice is preferred by the food processing industry for applications such as parboiling and canning [21]. 

Most studies regarding the genes associated with functional and sensory properties utilize rice across a wide range of amylose contents (0–30%). However, there is additional variation for functional and sensory properties of rice within the same amylose class. High amylose rice can be categorized using the single nucleotide polymorphisms (SNPs) of *Waxy* intron 1 and exon 6 [22,23]. Within the high amylose class, there are two major functional nucleotide polymorphisms in *Waxy* and *SSIIa*. The *Waxy* 10 SNP is associated with a specific RVA profile related to properties preferred by the food processing industry and it is also associated with gel consistency, an assay predicting hardness of the cooked rice [24,25,26,27]. The *SSIIa* is a key gene regulating gelatinization temperature (GelT) of rice; however, knowledge of its effect on viscosity and texture properties is lacking [28,29]. The interaction of these two genes (*Waxy* and *SSIIa*) on RS and functional and sensory properties within the high amylose genetic background has not been well investigated.

Since a near 2-fold difference in RS content was reported among high amylose varieties, there is opportunity to breed rice cultivars with enhanced nutritional properties associated with RS [11,13]. *Waxy, SSIIa* and two other loci reportedly regulate RS content in rice [11,12]. The structure of RSIII is formed after starch gelatinization during cooking. Double helixes of starch molecules that have a degree of polymerization of 10 or more realign during retrogradation as the starch cools [8]. These structures are dense with few associated water molecules [8]. We hypothesize that among high amylose rice varieties, differences in RS levels have different functional and textural properties.

Globally, there is increasing incidence of Type II diabetes, obesity, heart disease, and cancer. Several studies showed that rice consumption from regions that prefer low amylose rice were linked to increasing risk of developing Type II diabetes (in review article [16]). Thus, there is need to develop high RS rice that has texture and quality traits that suit preferences of consumers from different regions. Understanding the genetic control of RS and the impact on texture and functional quality within high amylose rice might lead to that goal.

The objectives of this study were to determine the combined effects of *Waxy* and *SSIIa* SNPs on RS content of rice in the high amylose class and the associated influence on the RVA viscosity profile and texture properties of cooked rice. It was concluded that natural genetic variation exits that can be used through breeding to increase RS content within high amylose haplotypes with minimum impact on processing quality and cooked rice texture.

## 2. Materials and Methods

### 2.1. Rice Varieties

Ten high amylose varieties were selected for this study along with one intermediate amylose variety, Wells, and one low amylose variety, Hidalgo (HDLG). These high amylose varieties were classified into three haplotypes-T-TT, T-GC and C-GC based on the T/C single nucleotide polymorphism (SNP) of *Waxy* exon 10 and TT/GC dinucleotide polymorphism (DNP) of *SSIIa* exon 8 [23,30]. Varieties having the T-TT haplotype are Dalidao (DA) and Jaya (JA), those with the T-GC haplotype are KN-1 B-361-BLK-2 (KN) and Dixiebelle (DXBL), those with the C-GC haplotype are CNTLR80076-44-1-1-1 (CN), Ghati Kamma Nangarhar (GK), Ghoal Champa (GC), Santa Julia (SJ), Tsipala 421 (TS), and L-202 (L202). The T/C SNP of *Waxy* exon 10 distinguishes between rice varieties having strong (T) versus weak (C) RVA viscosity curves [24]. The TT/GC DNP of *SSIIa* discriminates low (TT) and intermediate/high (GC) GelT of rice [28]. 

These 12 rice varieties were grown in 2016 at Stuttgart, AR and Rayne, LA using non-replicated plots and common cultural management practices for these locations. Rough rice samples were hand harvested at approximately 20% moisture and dried to 12%. Whole milled kernels (head rice) were obtained from milling 100 g aliquots of rough rice using a laboratory rice mill model PAZ-1 DTA (Zaccaria USA, Anna, TX, USA). A total of 2000 g of whole milled kernels from each location were sub-sampled for descriptive sensory analysis. A portion of the whole milled kernels were ground to flour and then passed through a 0.5 mm screen using a Cyclone Sample Mill (UDY Corp., Fort Collins, CO, USA) for grain quality analyses. All physicochemical and sensory analyses for each sample were determined in duplicate (12 varieties × 2 locations × 2 replicates).

### 2.2. Apparent Amylose Content

Apparent amylose content (AAC) of the milled rice flour was determined using the iodine spectrophotometric method of AACC 61-03 (American Association of Cereal Chemists) (1999). Briefly, the rice solution (samples or standard) was prepared to 1 mg/mL by sequentially adding 100 mg flour, 1 mL of 95% ethanol and 9 mL of 1N NaOH into a 100 mL volumetric flask to dissolve the starch for 24 h. This was followed by adding deionized water to 100 mL. The iodine color reaction was then performed by adding sequentially: 0.5 mL of rice flour solution, 5 mL water, 0.1 mL 1N acetic acid, 0.2 mL iodine solution (0.2% iodine in 2.0% KI) and then 4.2 mL of water. After mixing, the color developed was read at 620 nm. The AAC was determined against a standard curve of four calibrated rice standards as described previously reported [13]. The AAC was expressed on an ‘as is’ moisture basis.

### 2.3. Protein

The protein content in milled rice flour was determined using a CHNS/O Elemental Analyzer (2400 Series II CHNS/O Elemental Analyzer, PerkinElmer Health Sciences Inc., Shelton, CT, USA). The rice flour was first de-fatted with petroleum ether using a Soxtex 2050 auto extraction unit (Foss North America, Eden Prairie, MN, USA) as described in Bett-Garber et al. [31]. Briefly, 3 g of rice flour was weighed in a thimble and submerged into an extraction cup, which was filled with 70 mL of petroleum ether and then was heated to 135 °C for 20 min. Then the rice flour was raised out of the solvent and was rinsed for 30 min from the condensed petroleum ether at a flowrate of 3 to 5 drops/s (as recommended by Foss Inc.). After extraction, the remaining petroleum ether was removed via evaporation, and the flour was then dried under vacuum before nitrogen analysis. For analysis, 1.5 to 2.5 mg was weighed and placed into a capsule and combusted in a pure oxygen environment. The nitrogen content in the combustion gases was determined following the manual of the Elemental Analyzer. A factor of 5.95 was used to convert nitrogen to protein (%, dry wt.).

### 2.4. Pasting Properties

The flour pasting properties were determined using a Rapid Visco Analyser (RVA) (Newport Scientific, model RVA-4, Perten Instrument, Huddinge, Sweden) following the AACC 61-02 method (1999). A 3 g sample of flour was weighed and added to a canister containing 25 g of deionized water and then mixed with a paddle. The flour/water mixture was subjected to a heat-hold-cool temperature cycle and the pasting properties of the curve were determined. The time (min:s)/temperature cycle is as follows: 0:00–1:00/50 °C, 1:00–4:45/50 °C–95 °C linearly (heating cycle), 4:45–7:15/95 °C (holding cycle), 7:15–11:06/95 °C–50 °C linearly (cooling cycle), and 11:06–12:30/50 °C. The paste viscosity parameters for the RVA curve were peak viscosity (PV), hotpaste viscosity (HPV), coolpaste viscosity (CPV), breakdown (BD), setback from PV (SBfPV = CPV − PV), and setback from HPV (SBfHPV = CPV − HPV). PV is the maximum viscosity recorded during the heating and holding cycles. HPV is the minimum viscosity after PV. CPV is the viscosity achieved at the end of the test. BD is PV − HPV.

### 2.5. Thermal Properties

The thermal properties of milled rice were determined using a differential scanning calorimeter (DSC6, Perkin-Elmer Corp., Norwalk, CT, USA). A sample of 8 mg of rice flour (defatted with petroleum ether) was weighed into an aluminum pan and 16 mg of deionized water was added. The pan was hermetically sealed and equilibrated at room temperature for 1h before being heated at a rate of 5 °C/min from 30 to 100 °C. The reference was a sealed empty pan. The peak temperature (Tp, °C) representing the GelT and the enthalpy of gelatinization (∆H, joule/g) were calculated using the associated software (Pyris series, Thermal Analysis Manager Suite N537-0605 Version 4.0, Perkin-Elmer Corp.).

### 2.6. Rice Cooking Protocol

Two of the high amylose rice varieties with differing viscosity profiles (SJ and DXBL) that were harvested in AR were subjected to preliminary cooking and sensory tests using both 1:2 and 1:1.8 (*w*/*w*) rice to water ratios. Both rice varieties that were cooked at 1:1.8 rice to water ratio were fully hydrated and remained intact, unlike those cooked at 1:2 ratio. Thus, the 1:1.8 (*w*/*w*) rice to water ratio was used for this study.

A portion of milled rice (600 g) was rinsed and strained three times, placed in the rice cooker insert, and then water was added to reach the final weight of 1680 g of rice plus water. The rice samples were cooked in a rice cooker (Aroma ARC-787D-1NG, San Diego, CA, USA) at a temperature of 100 °C, until automatically stopped, indicating all water had been absorbed.

### 2.7. Sensory Evaluation

Thirteen panelists who were previously trained in the principles and concepts of descriptive sensory analysis participated in the study [18,31,32]. The lexicon for the descriptive rice texture attributes, the definition and the intensity reference samples used by the panel are presented in Appendix A. A total of 14 texture attributes were used for describing texture during four phases of eating the rice-evaluating the cooked rice kernels with tongue before chewing, at first bite, during chewing, and after swallowing. The descriptive lexicon and evaluation procedure used was as described in Champagne et al. [33] with some modification. In brief, there was a total of 24 samples and each sample was coded and presented to the panelists twice in a randomized design (12 varieties × 2 locations × 2 replicates). Each sensory session included the evaluation of four samples and a standard. The standard is a commercial rice (Mahatma XLG, Riviana Foods, Inc., Houston, TX, USA) purchased from a local store and was presented at the beginning of each session as a preparation sample to calibrate the panel. All rice samples were left in the rice cooker for 10 min in a warming state after the automatic shut-off of the rice cooker prior to presentation to the panelists. Between samples distilled water was used by the panelists to rinse their mouths. 

### 2.8. Resistant Starch 

The concentration of resistant starch (RS% dw) in the cooked rice was determined based on the AOAC Method 2,002,02 (AOAC International, 2007) using a Resistant Starch Assay kit from Megazyme (K-RSTAR 08/11, Wicklow, Ireland) and expressed as % resistant starch/cooked rice (dry wt basis) as described in Chen et al. [13]. Briefly, the freshly cooked rice was minced and then a 0.5 g portion was weighed and incubated in a shaking water bath with pancreatic α-amylase and amyloglucosidase for 16 h at 37 °C. An equal volume of ethanol was added to the digest to precipitate the RS. After centrifugation, the pellet was washed twice in 50% *v/v* ethanol. The RS in the pellet was dissolved in 2 M KOH, neutralized with acetate buffer and hydrolyzed to glucose with amyloglucosidase. After adding glucose oxidase/peroxidase reagent to the samples the absorbance of each solution was determined using a spectrophotometer at 510 nm. The glucose concentration was then converted to RS.

### 2.9. Statistical Analyses

All statistical analyses were conducted using JMP software (Version 14, SAS Institute, Cary, NC, USA). Tukey–Kramer HSD was used for mean comparisons between all traits. The relationship of grain quality traits with RVA parameters and with sensory attributes were analyzed using pairwise correlation coefficient and Principle Component Analysis (PCA). Discriminant analysis, using stepwise covariance analysis and *p* value forward variable selection (*p* < 0.05), was performed to determine the variables of RVA curves and texture attributes that were best in discriminating the three *Waxy*-*SSIIa* haplotypes. 

## 3. Results

### 3.1. AAC, RS and Protein Content, Tp and ∆H

The means and ranges of AAC, RS, protein, Tp and ∆H of the three high amylose haplotypes (Table 1), T-TT, T-GC and C-GC are presented in Table 2. Location means of these traits for individual varieties are provided in Appendix A. Across the three high AAC haplotypes, mean comparison showed that all traits were not significantly different except Tp with T-TT haplotype having a lower Tp than T-GC and C-GC. For each of these traits, the value ranges were as follows: protein (53–81 g/kg), AAC (252–292 g/kg), RS (27–45 g/kg), Tp (66.6–78.8 °C), and ∆H (7.6–11.0 J/g). Wells, an intermediate amylose cultivar, had comparable protein content and ∆H to all three high amylose haplotypes, but it had lower AAC and RS contents. Hidalgo, a low amylose cultivar, also had comparable protein content, but lower AAC and RS contents and higher Tp and ∆H, relative to the three high amylose haplotypes.

The highest RS found in our study was higher than the highest one reported in Bao et al. [11] (range = 0.3–24.2 g/kg covering low to high amylose rice varieties) and in Parween et al. [12] (range = 1.7–33.7 g/kg). The RS of high amylose rice varieties in our previous report ranged from 23.7 to 45.5 g/kg [13]. The highest AAC was 274 g/kg, which was higher than a previous report from our lab of 247 g/kg [22] and was due to the inclusion of some varieties that were high in AAC and RS that we had identified previously [13]. The protein levels in our study were in a lower range compared to what was reported in Champagne et al. [18] (59–112 g/kg) although Tp values were within the range of GelT that they had reported (62–78 °C) (HDGL, a low-amylose high GelT variety, was excluded).

### 3.2. RVA Viscosity Parameters

The means and ranges of RVA viscosity parameters of the high amylose haplotypes, Wells, and Hidalgo are presented in Table 3; while location means of these traits for individual varieties are provided in Appendix A. Haplotypes T-TT and T-GC were not significantly different in PV, CPV, SBfHPV and SBfPV, while T-TT had higher HPV and lower BD than T-GC indicating potential impact of the DNP of the *SSIIa* allele. Both T-TT and T-GC had higher PV, HPV, CPV and SBfHPV than C-GC which supports the reported association of the T allele of the *Waxy* exon 10 SNP with a strong viscosity curve and the C allele with a weak viscosity curve [24,26].

The viscosity of RVA parameters in this report were within the ranges of those in [13] except PV of C-GC (the Weak_Int haplotype in [13]), of which the minimum PV of current study was 15 RVU lower. Comparing the RVA viscosity in [26], most of the absolute RVA parameters (PV and HPV) had the viscosity values within the ranges of those reported in [26] except the minimum CPV, which was approximately 25 RVU lower in the current study.

### 3.3. Association of Grain Quality Traits with RVA Parameters

Bivariate correlation analyses of grain quality traits and RVA parameters were conducted, either including all high amylose varieties, or grouping varieties based on T/C SNPs of *Waxy* exon 10 (Appendix A). RS was highly correlated with AAC (*r* = 0.88–0.89), either when all high amylose varieties were included or when varieties were grouped based on *Waxy* exon 10 SNPs. RS was correlated with SBfHPV across all high amylose varieties (*r* = −0.46) as well as with rice varieties of *Waxy* exon 10 T− and C-SNP types (*r* = −0.93 and −0.76, respectively).

Thermal property of rice appeared to affect the RVA curve. When all high amylose varieties were included, Tp was negatively correlated with all RVA parameters except BD and SBfHPV. For varieties with the T-TT and T-GC haplotypes, Tp was negatively correlated with HPV and positively correlated with BD. For varieties of C-GC haplotype, Tp was negatively correlated with BD and positively correlated with FV, SBfHPV and SBfPV, and ∆H was positively correlated with HPV, FV and SBfPV.

PCA analysis of grain quality traits and RVA parameters of high amylose varieties is presented in Table 4 and Figure 1. The first three PCAs explained 82.3% of the total variance with PC1, PC2 and PC3 accounting for 42.3%, 23.7% and 16.4% (all with eigenvalue > 1). PC1 included Tp (negative loading) and all RVA parameters except BD (all positive loadings). PC2 included (eigenvector > |0.25|) grain quality traits of RS, AAC and Protein (positive loadings) along with Tp and SBfHPV (negative loadings), whereas RS had the highest loading (0.58). PC3 was primarily loaded with BD. Score plot of PC1 and PC2 (Figure 1a) shows that varieties of T-GC and T-TT haplotypes were positioned on the positive scale of PC1, while varieties of C-GC were located on the negative scale of PC1. Between T-GC and T-TT, varieties of T-TT occupied on higher positive scale of PC1 than varieties of T-GC. The score plot also shows that varieties with higher RS were positioned on the positive scale of PC2, while varieties of lower RS were distributed on the negative scale of PC2. Thus, higher loading variables represented in PC1 were those associated with genetics of *Waxy* and *SSIIa*, while those in PC2 were those that were primarily associated with RS, not with the *Waxy* and *SSIIa*. The variance of SBfHPV was distributed between PC1 and PC2.

### 3.4. Texture Attributes 

Fourteen texture attributes of the cooked rice were evaluated in four chewing phases (Appendix A). The means and the ranges of three high amylose haplotypes along with Wells and Hidalgo are presented in Table 5 and Table 6 and Figure 2, and the location means of these attributes for individual varieties are provided in Appendix A. Between T-TT and T-GC, no statistically significant differences in texture attributes were found. Comparison between T-GC and C-GC haplotypes found that C-GC had higher Starch, StickGn, CohevM, while it had lower IntactP and Chews. Comparing Wells with the three high amylose haplotypes, Wells had eight, six and four texture attributes different from T-TT, T-GC and C-GC haplotypes, respectively. The traits that differed the most between Wells and high amylose groups were in Phase I, then Phase II and III, and none in Phase IV. The four attributes that C-GC haplotype differed from those of Wells were Slick (lower), StickLp (lower) and Rough (higher) in phase I, and UniB (lower) in phase III.

The low amylose variety, Hidalgo, had most of the attributes different from the high amylose haplotypes with only one, three and two attributes that were not significantly different from T-TT, T-GC and C-GC haplotypes, respectively (Table 5 and Table 6). Moist was the only trait that was similar across all three haplotypes, Wells and Hidalgo. 

The comparison of the current study results with other references for textural properties of cooked rice was not possible because the rice-to-water ratios differ across studies. This ratio affects cooked rice texture [33]. The rice-to-water ratios were variable in [18] and was 1:1.7 (*w/w*) in [31,33] for the high amylose class-type.

### 3.5. Association of Grain Quality Traits with Texture Attributes

Pairwise correlation of grain quality traits with the texture attributes is presented in Appendix A. RS and protein were positively (*r* = 0.47 and 0.49, respectively, both at *p* < 0.05) and Tp was negatively correlated with Rough (*r* = −0.59, *p* < 0.01). Protein was also positively correlated with StickGn (*r* = 0.45, *p* < 0.05). Tp was also positively correlated with Starch, StickLp and negatively with residuals (*r* = 0.52 at *p* < 0.05; *r* = 0.65 at *p* < 0.01; *r* = −0.45 at *p* < 0.05, respectively). Pairwise correlation among texture attributes of high amylose varieties showed that some attributes were highly correlated with others, specifically, Starch, CohevM, IntactP, Hard, and Cohev.

PCA analysis of grain quality traits and texture attributes of high amylose varieties are presented in Table 7 and Figure 3. The first three PCAs accounted for a total of 63.3% variance with PC1, PC2 and PC3 having 36.6%, 15.9% and 10.9%, respectively. PC1 included (eigenvector > |0.25|) positive loadings of Starch, StickLp, Cohev, CohevM, and Tp and negative loadings of Hard, IntactP, and Chews. PC2 included (eigenvector > 0.35) RS, AAC, protein, Rough and StickGn. PC3 was primarily loaded with two variables, ∆H and Slick. The score plot (Figure 3a) shows PC1 divides high amylose varieties based on the *Waxy* exon 10 SNP in that C-GC varieties are clustered on the positive scale of PC1 and T-TT and T-CC varieties are clustered on the negative scale of PC1, while PC2 divided varieties having higher and lower RS. StickGn and Rough were two texture attributes that had higher loading for PC2 compared to PC1.

### 3.6. Discriminant Analysis

Discriminant analysis identified two sets of RVA parameters, PV plus BD (both had *p* < 0.0001) and PV plus HPV (both had *p* < 0.0001), that were best in discriminating between the three high amylose haplotypes (Figure 4a).

For the texture attributes of the high amylose varieties, discriminant analysis found that the Starch (*p* < 0.0001), IntactP (*p* < 0.01), and Rough (*p* < 0.05) variables were associated with the three haplotypes except two samples, one is a T-TT from LA being grouped to T-GC and the other one is a T-GC from AR being grouped to T-TT (Figure 4b).

## 4. Discussion

Consumption of rice with higher levels of RS could help to reduce the incidence of type two diabetes. Progress towards the development of cultivars with high RS requires a better understanding of the genetics controlling RS levels and the physiochemical and sensory properties associated with this trait.

In the present study, the three high AAC haplotypes studied had the same functional SNPs for intron 1 (G) and exon 6 (A) in *Waxy*, which was reflected in them having similar AAC as reported previously [22]. RS was highly correlated with AAC agreeing with the hypothesis that RS is primarily formed from retrograded amylose [7]. Bao et al. [11] included rice varieties of low, intermediate and high AAC types and found RS was highly associated with AAC (*R*^2^ = 0.75). On the other hand, Parween et al. [12] reported a low correlation of RS with amylose content (*r* = 0.42). Growing environment, genotype selection and cooking method all might affect the RS content. As reported in [12] the heritability of the RS data was low (*H*^2^ = 0.4) from three replications of growth in the wet seasons. In Chen et al. [13] the high amylose varieties grown in TX had lower RS (32.0 g/kg) than those grown in AR (34.4 and 34.9 g/kg on average for two different years).

Haplotype T-TT had lower Tp than T-GC and C-GC, which agrees with previous reports that the functional DNP of *SSIIa* is related to the enzyme activity of *SSIIa* and that the GC allele has higher enzyme activity, while the TT allele has zero to low activity. Thus, rice varieties with the GC DNP have longer exterior B chains and higher Tp [28,29]. Our study showed that the RS was not significantly different across the three high amylose haplotypes, which agreed with a previous report [13]. Both studies used rice varieties with high amylose content and included both DNP genetic variants of the *SSIIa* gene. On the other hand, other reports that utilized rice varieties across a wide range of amylose contents found that RS was associated with *SSIIa* along with *Waxy* and other starch synthesis related genes [11,12]. Bao et al. [11] identified four candidate genes associated with RS, from high to low *R*^2^: ADP-*glucose pyrophosphorylase small subunit 1*, *isoamylase 1*, *Waxy*, and *SSIIa*, while Parween et al. [12] found RS was primarily associated with *SSIIa* gene.

Both the *Waxy* exon 10 SNP and *SSIIa* exon 8 DNP were associated with RVA viscosity profiles. Varieties having the *Waxy* exon 10 T SNP had higher values for PV, HPV, CPV and SBfHPV compared to those with the C SNP. This agrees with the study of Traore et al. [24] where it was shown that the differences were related to the proportions of soluble versus insoluble amylose. As to *SSIIa*, previous reports using genotypes across the low, intermediate and high amylose classes, found that *SSIIa* was a minor player in controlling RVA traits, while *Waxy* had a major effect on these functional properties [35,36]. This is the first report indicating that among only high amylose varieties, *SSIIa* plays a critical role in controlling RVA traits by affecting HPV and BD. Plus, linear discriminant analysis identified that the RVA profiles of the three high amylose haplotypes could be best represented by just two variables, PV and HPV.

HPV and BD are the viscosity parameters that determine the mechanical shear strength of starch granules at a holding temperature of 95 °C and they are associated with swelling and rigidity of starch granules, protein and amylose contents, and amylopectin structures [37]. In the current study, the PV was not different between T-TT and T-GC, thus the BD difference between the two haplotypes was due to the difference in HPV. At 95 °C holding temperature of the RVA thermal curve, the crystalline regions of the exterior chains should have melted (GetT < 95 °C), thus the HPV might reflect the amylopectin chains other than those with a double helical structure. Previous study showed that the T-TT haplotype, which had higher HPV and lower Tp, had more shorter chains compared to those with T-GC [34]. The short amylopectin chains were reported to increase swelling, while long chains may prevent swelling [38]. Short amylopectin chains that belong to the internal structure of amylopectin were reportedly correlated with higher swelling power and lower amylose leaching during heating [39].

The T-GC haplotype is preferred by the U.S. rice processing industry because kernels maintain their integrity after parboiling and canning and there is less soluble amylose loss compared to the C-GC haplotype [21,24]. Soluble amylose levels reportedly are negatively correlated with all five RVA parameters [24]. Pairwise correlation analysis indicated that RS was negatively associated with SBfHPV but not with the other RVA parameters. Plus, PCA analysis showed that RS was grouped in PC2 and had the highest loading among all traits and SBfHPV was the only RVA parameter in PC2 with an eigenvector ≥ |0.25|; while PC1 was associated with traits (including most of the RVA parameters–PV, HPV, FV, SBfHPV and SBfPV) linked with the *Waxy* exon 10 allele according to score and loading. Therefore, high amylose varieties with increased RS may have processing quality similar to their lower RS counterparts. Further study of the parboil processing quality of these varieties would be needed to confirm this. 

SBfHPV measures the amount of retrograded starch molecules created during the cooling phase of an RVA analysis [21]. RS is a compact crystalline structure formed from the realignment of the retrograded, double helices of amylose and has low water-holding capacity [8]. Thus, the negative correlation between RS and SBfHPV suggests that a portion of the retrograded amylose that forms RS binds less water and thus reduces the rise in viscosity from HPV to CPV. 

Other starch characteristics may also explain some of the variation in the RVA curve for the high amylose varieties. By dividing the high amylose varieties by the T/C SNP of *Waxy* exon 10 into two sub-groups, Tp differed in correlations with other RVA parameters. For those having the T SNP, Tp was negatively correlated with HPV and positively correlated with BD and that was due to having varieties of both the TT and GC DNP of *SSIIa* in this sub-group. For the varieties with the C-GC haplotype, Tp was negatively correlated with BD and positively with FV, SBfHPV and SBfPV. In addition to the double helical structure of exterior chains of amylopectin, other starch structures, i.e., the medium chains of amylopectin and amylose as well as the amylopectin internal structure, were reported to be associated with GelT and pasting properties of high amylose rice varieties [39,40].

High amylose rice varieties that differ in cooked rice hardness have been identified by rice breeding programs using the gel consistency and RVA methods [24,25]. The associations of gel consistency and cooked rice hardness with the T/C SNP of *Waxy* exon 10 were recently confirmed by Tran et al. [25]. However, hardness is not the only factor determining palatability of the cooked rice [27,41]. This report expanded the knowledge of the genetics controlling cooked rice texture beyond hardness by studying the combined effects of *SSIIa* and *Waxy* on 14 texture attributes of cooked rice. Most of the sensory attributes of cooked rice between T-TT and T-GC were similar except that T-TT had lower Slick and StickGn and higher Rough than T-GC, all in chewing phase I. On the other hand, five attributes were significantly different between T-GC and C-GC across all chewing phases. This suggests that the sensory attributes in these high amylose varieties were impacted more by the soluble/insoluble amylose fractions than by amylopectin structures. However, pairwise analysis showed some attributes were highly correlated. Using linear discriminant analysis we identified three texture attributes (i.e., Starch, IntactP and Rough) that best differentiate the three *Waxy* and *SSIIa* haplotypes with 90% accuracy. Rice with intermediate amylose is preferred by many consumers in the U.S., Europe and in some parts of China, South America and south Asia [19]. The high amylose haplotype C-GC had only four texture attributes that differed from the intermediate amylose variety Wells, while T-GC and T-TT differed by six and eight attributes, respectively. Thus, from the prospect of consuming rice with more health benefits, i.e., lower GI and higher RS, the C-GC high amylose haplotype may be more acceptable to consumers that prefer intermediate amylose type of rice, although it is less desirable for the processing industry.

Within the high amylose class, >2 folds difference in RS content have been reported [11,13]. Thus, these varieties can be used to enhance RS using traditional plant breeding techniques. However, consumer acceptance is crucial for the success of introducing high RS rice as a healthier alternative into the marketplace. The score plot of PCA analysis showed PC1 divided varieties based on *Waxy* exon 10 SNPs, those with C on positive scale of PC1, those with Ton negative scale of PC1; thus, suggesting that higher loading traits of grain quality and sensory attributes in PC1 were more likely linked to differences in the *Waxy* exon 10. The score plot of PC2 divided high RS and low RS varieties, thus suggesting higher loading traits in PC2 might be related to RS plus protein, AAC, Rough and StickGn. Pairwise analysis further confirmed that for cooked rice texture, Rough was impacted by RS, and protein was correlated with Rough and StickGn.

Positive correlations of protein with roughness were reported by Champagne et al. [42,43]. The correlation coefficient was low, but significant statistically, which agreed with our current finding. The current study is the first report demonstrating that RS is associated with the roughness of cooked rice texture within high amylose varieties. Previous descriptive sensory analysis evaluated premium and lower (second best) quality pairs of rice varieties from each of nine countries [18]. Rough along with Slick and Springe were the major traits that distinguished the textural differences between these pairs but consumer preferences differed across global regions [18]. For example, of the nine countries that participated in the study, Brazil, Iran, and Pakistan had premium rice-types with higher roughness than the second-best type, suggesting that roughness might be one of the cooked rice texture attributes preferred by their consumers [18]. This study demonstrates that availability of rice varieties with elevated RS that, based on the *Waxy* and *SSIIa* haplotypes, have different textural attributes that may be suitable for different consumer markets in various parts of the world. 

The evidence indicates that high amylose varieties would be suitable for use in food products designed to have a lower impact on blood glucose levels. This is the first demonstration of native rice with high resistant starch that has sensory properties suitable for use as cooked rice, as opposed to previous reports that have created high resistant levels using biotechnology tools [44]. 

## 5. Conclusions

In conclusion, high amylose rice varieties with >1.6-fold variation in RS were studied to better understand the effects of genetics and RS on sensory and RVA pasting properties. Three haplotypes were studied that combined functional genetic variants of *Waxy* and *SSIIa* of high amylose varieties and the key pasting parameters and texture attributes were identified. RS content had little association with paste viscosity indicating that there would be no expected effect on processing quality that is important to the parboiling industry which prefers high amylose rice. The amount of RS was significantly related to one texture attribute, Rough, out of the 14 studied. However, among high amylose rice haplotypes with elevated RS levels variation exists in sensory properties that can likely meet the divergent expectations for consumer acceptance in various rice consuming regions of the world. Rice with higher RS offers the rice industry an opportunity to capture a new market of consumers that desire foods with a low glycemic index. 

## Figures and Tables

**Figure 1 foods-11-00094-f001:**
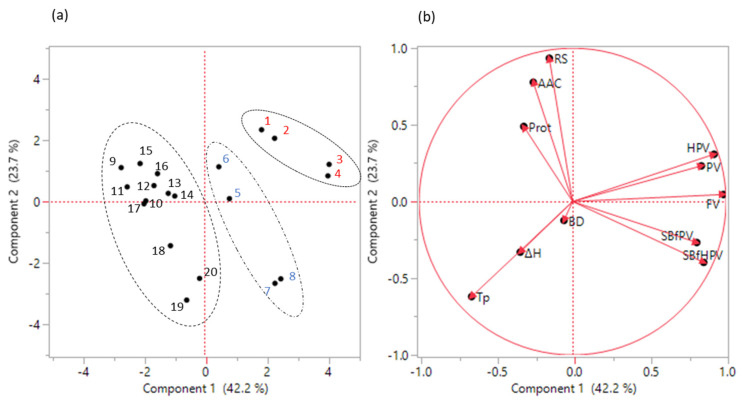
The score plot (**a**) and loading plot (**b**) of PCA analysis of grain functional traits and RVA parameters of 10 high AAC varieties grown in LA and AR. In (**a**), the numbers in red font (No. 1–4) are varieties in the T-TT haplotype, blue font (No. 5–8) are the T-GC haplotype, and black font (No. 9–20) are the C-GC haplotype. For variety names and growing states for these numbers, refer to Table 1. In (**b**), for trait abbreviations see Table 2 and Table 3.

**Figure 2 foods-11-00094-f002:**
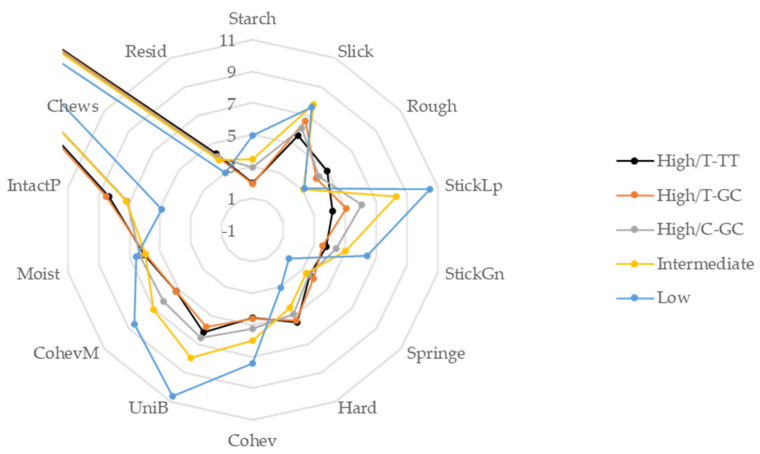
Spider graph of 14 texture attributes of three high amylose haplotypes (High/T-TT, High/T-GC, High/C-GC), intermediate amylose and low amylose types. Values for Chews are 32.8, 33.2, 30.8, 30.7 and 23.0 for High/T-TT, High/T-GC, High/C-GC, Intermediate and Low, respectively.

**Figure 3 foods-11-00094-f003:**
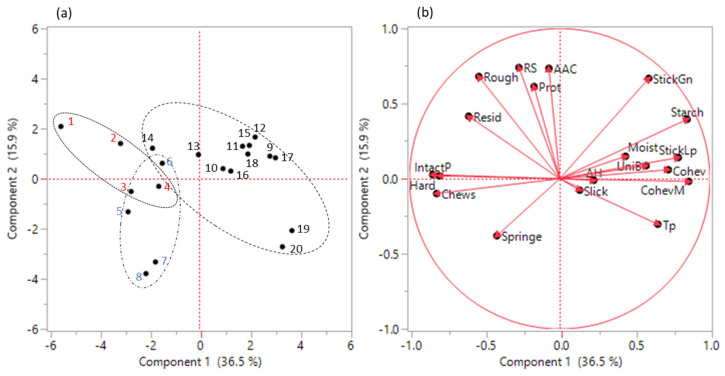
The score plot (**a**) and loading plot (B) of PCA analysis of grain quality traits and texture attributes of 10 high AAC varieties grown in LA and AR. In (**a**), the numbers in red font (No. 1–4) are varieties in the T-TT haplotype, blue font (No. 5–8) the T-GC haplotype, and black font (No. 9–20) the C-GC haplotype. For variety names and growing states for these numbers, refer to Table 1. In (**b**), for trait abbreviations see Table 2 and Appendix A.

**Figure 4 foods-11-00094-f004:**
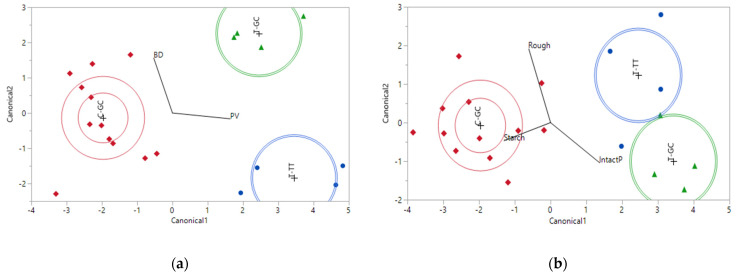
Linear discriminant analysis of RVA parameters (**a**) and texture attributes (**b**) in association with three high amylose haplotypes.

**Table 1 foods-11-00094-t001:** Rice varieties and their predicted classes of apparent amylose, viscosity and gel temperature (GelT) based on *Waxy* exon 10 single nucleotide polymorphism (SNP) and *SSIIa* exon 8 dinucleotide polymorphism (DNP).

Variety ^†^	Abbreviation	AAC ^‡^	Viscosity ^§^	GelT ^¶^	High AAC Haplotype (*Waxy* ex10 T/C SNP-*SSIIa* Exon 8 TT/GC DNP)	Associate Numbers Listed in Figure 1 and Figure 3
Dalidao	DA	High	Strong	Low	T-TT	1-AR, 2-LA
Jaya	JA	High	Strong	Low	T-TT	3-AR, 4-LA
KN-1 B-361-BLK-2	KN	High	Strong	Int	T-GC	5-AR, 6-LA
**Dixiebelle**	DXBL	High	Strong	Int	T-GC	7-AR, 8-LA
CNTLR80076-44-1-1-1	CN	High	Weak	Int	C-GC	9-AR, 10-LA
Ghati Kamma Nangarhar	GK	High	Weak	Int	C-GC	11-AR, 12-LA
Ghoal Champa	GC	High	Weak	Int	C-GC	13-AR, 14-LA
Santa Julia	SJ	High	Weak	Int	C-GC	15-AR, 16-LA
Tsipala 421	TS	High	Weak	Int	C-GC	17-AR, 18-LA
**L-202**	L202	High	Weak	Int	C-GC	19-AR, 20-LA
Wells	Wells	Int	IntAAC	Int	NA	NA
Hidalgo	HDLG	Low	LowAAC	high	NA	NA

^†^ Samples in bold font are high amylose U.S. varieties. ^‡^ AAC, apparent amylose content classes were predicted by SNPs in Intron 1 and Exon 6 of the *Waxy* gene [22]. ^§^ Strong and Weak RVA classes among high amylose varieties were classified based on SNP in Exon 10 of the *Waxy* gene [26]. ^¶^ GelT, gelatinization temperature classes determined based on SNPs in Exon 8 of the *SSIIa* gene which distinguishes low from intermediate (Int) and high GT classes [28]. Within Int/High GT class, the low AAC variety has high GT [34].

**Table 2 foods-11-00094-t002:** Protein, AAC, RS content, and thermal properties of all rice varieties ^†^.

AAC Class	High AAC Haplotype and Other Varieties		Protein (g/kg)	AAC (g/kg)	RS (g/kg)	Tp (°C)	∆H (J/g)
High	T-TT	mean	68A	277A	41A	67.6C	8.2B
	T-GC	mean	63A	268A	36A	77.6B	9.2B
	C-GC	mean	68A	276A	38A	77.6B	9.3B
Intermediate	Wells	mean	61A	215B	19B	77.6B	8.5B
Low	Hidalgo	mean	62A	108C	2C	82.3A	11.6A
High	T-TT	(range)	(56–81)	(271–282)	(39–45)	(66.6–68.8)	(7.6–8.9)
	T-GC	(range)	(60–66)	(253–287)	(28–45)	(76.4–78.8)	(8.6–10.2)
	C-GC	(range)	(53–80)	(252–292)	(27–44)	(76.7–78.6)	(7.9–11.0)
Intermediate	Wells	(range)	(57–65)	(214–216)	(18–20)	(77.3–78.0)	(8.5–8.5)
Low	Hidalgo	(range)	(61–64)	(107–110)	(2–2)	(82.2–82.4)	(11.5–11.7)

^†^ Different capital letters for the means indicate significant difference between high amylose haplotypes and intermediate and low AAC varieties for each trait at *α* = 0.05 using Tukey HSD’s test. Prot., protein; AAC, apparent amylose content; RS, resistant starch.

**Table 3 foods-11-00094-t003:** Viscosity properties of all rice varieties ^†^.

AAC Class	High AAC Haplotype and Other Varieties		PV (RVU)	HPV (RVU)	BD (RVU)	CPV (RVU)	SBfHPV (RVU)	SBfPV (RVU)
High	T-TT	mean	263B	197A	66C	361A	164A	99A
	T-GC	mean	269B	156B	114B	335A	179A	66AB
	C-GC	mean	186C	107C	80C	240B	134B	54B
Intermediate	Wells	mean	255B	128BC	127B	252B	124BC	−3C
Low	Hidalgo	mean	319A	136BC	183A	221B	85C	−98.1D
High	T-TT	(range)	(237–287)	(178–216)	(59–71)	(326–394)	(148–178)	(90–108)
	T-GC	(range)	(257–292)	(146–171)	(109–121)	(314–354)	(155–208)	(34–95)
	C-GC	(range)	(153–208)	(84–136)	(52–102)	(202–275)	(118–151)	(24–95)
Intermediate	Wells	(range)	(242–267)	(124–132)	(117–136)	(246–258)	(121–126)	(−9–4)
Low	Hidalgo	(range)	(304–334)	(134–138)	(170–196)	(216–225)	(82–87)	(−109–88)

^†^ Different capital letters for the means indicate significant difference between high amylose haplotypes and intermediate and low AAC varieties for each trait at α = 0.05 using Tukey HSD’s test. PV, peak viscosity; HPV, hotpaste viscosity; BD, breakdown viscosity = (PV − HPV); CPV, coolpaste viscosity; SBfHPV = CPV − HPV; SBfPV = CPV − PV.

**Table 4 foods-11-00094-t004:** PCA ^†^ analysis of traits of grain quality and RVA parameters ^‡^.

Traits	PC1	PC2	PC3
RS	−0.07	**0.58**	0.15
AAC	−0.12	**0.48**	0.23
Protein	−0.15	**0.30**	**−0.26**
Tp	**−0.31**	**−0.38**	**0.26**
∆H	−0.16	−0.20	0.21
PV	**0.39**	0.14	**0.35**
HPV	**0.43**	0.19	0.03
BD	−0.03	−0.08	**0.68**
FV	**0.46**	0.03	0.09
SBfHPV	**0.40**	−0.25	0.17
SBfPV	**0.38**	−0.17	**−0.34**
*R*^2^ (%)	42.2	23.7	16.4

^†^ Bold fonts are those having eigenvectors > |0.25|. ^‡^ Abbreviation for the traits, see Table 2 and Table 3.

**Table 5 foods-11-00094-t005:** Cooked rice texture attributes of 3 high amylose haplotypes, Wells and Hidalgo ^†^.

AAC Class	High AAC Haplotype and Other Variety		Phase I				
			Starch	Slick	Rough	StickLp	StickGn
High	T-TT	mean	1.94C	5.58C	4.99A	4.19D	3.77CD
	T-GC	mean	1.91C	6.62ABC	4.17AB	5.09CD	3.52D
	C-GC	mean	2.92B	6.11BC	4.35A	6.07C	4.39BC
Intermediate	Wells	mean	3.46B	7.79A	3.11B	8.31B	5.00B
Low	Hidalgo	mean	4.93A	7.56AB	3.16B	10.51A	6.42A
High	T-TT	(range)	(1.8–2.1)	(5.1-6.6)	(4.2–5.7)	(3.5–4.6)	(3.3–4.1)
	T-GC	(range)	(1.7–2.3)	(5.8-7.7)	(3.6–4.7)	(4.6–6.0)	(3.2–4.2)
	C-GC	(range)	(2.5–3.4)	(4.7-6.7)	(3.6–5.1)	(5.0–7.5)	(3.8–4.8)
Intermediate	Wells	(range)	(3.1–3.8)	(7.1-8.5)	(2.5–3.7)	(8.1–8.5)	(4.6–5.4)
Low	Hidalgo	(range)	(4.8–5.0)	(6.8-8.3)	(3.0–3.3)	(10.3–10.7)	(6.4–6.5)

^†^ Different letters for the means within each column indicate significant differences between haplotypes, Wells, and Hidalgo at α = 0.05 using Tukey’s HSD test. Abbreviation for the texture attributes, see Appendix A.

**Table 6 foods-11-00094-t006:** Cooked rice texture attributes of 3 high amylose haplotypes, Wells and Hidalgo ^†^.

Phase II			Phase III			Phase IV		
Springe	Hard	Cohev	UniB	CohevM	Moist	IntactP	Chews	Resid
3.61A	5.47A	4.57D	6.15C	5.19C	6.05A	8.35A	32.79AB	4.34A
3.91A	5.37A	4.63CD	5.80C	5.19C	5.97A	8.49A	33.19A	4.13A
3.57A	4.88A	5.21BC	6.54C	6.20B	6.34A	7.15B	30.83B	4.10A
3.36A	4.44AB	5.98B	7.98B	7.04AB	5.95A	7.19AB	30.68AB	3.90A
1.95B	3.05B	7.41A	10.64A	8.55A	6.55A	4.93C	22.99C	2.98B
(3.1–3.9)	(5.1–5.8)	(4.3–4.9)	(5.3–7.1)	(4.9–5.5)	(5.8–6.4)	(8.0–8.8)	(31.3–34.4)	(3.9–4.8)
(3.5–4.3)	(5.2–5.6)	(4.2–4.9)	(5.4–6.4)	(4.8–5.6)	(5.6–6.4)	(8.1–8.9)	(32.1–34.4)	(4.0–4.2)
(3.1–3.8)	(4.3–5.7)	(4.3–5.9)	(5.9–7.0)	(4.7–6.9)	(5.7–7.3)	(6.4–8.0)	(28.7–32.9)	(3.8–4.5)
(3.0–3.7)	(4.2–4.7)	(5.9–6.0)	(7.8–8.1)	(6.8–7.3)	(5.9–6.0)	(6.7–7.7)	(29.8–31.6)	(3.8–4.0)
(1.4–2.5)	(1.9–4.2)	(7.2–7.6)	(10.0–11.3)	(8.0–9.1)	(6.5–6.6)	(4.4–5.5)	(22.0–24.0)	(2.8–3.1)

^†^ Different letters for the means within each column indicate significant differences between haplotypes, Wells, and Hidalgo at α = 0.05 using Tukey’s HSD test.

**Table 7 foods-11-00094-t007:** PCA analysis of traits of grain quality and texture attributes ^†^.

Traits ^‡^	PC1	PC2	PC3
RS	−0.10	**0.43**	0.06
AAC	−0.03	**0.42**	0.21
Prot	−0.06	**0.35**	−0.01
Tp	**0.25**	−0.17	**0.37**
∆H	0.09	−0.01	**0.55**
Starch	**0.32**	0.23	0.09
Slick	0.05	−0.04	**0.50**
Rough	−0.20	**0.39**	−0.11
StickLp	**0.30**	0.08	0.17
StickGn	0.23	**0.39**	0.12
Springe	−0.16	−0.22	0.16
Hard	**−0.30**	0.01	0.20
Cohev	**0.27**	0.03	−0.10
UniB	0.22	0.05	−0.16
CohevM	**0.33**	−0.01	−0.23
Moist	0.17	0.09	0.01
IntactP	**−0.32**	0.02	0.20
Chews	**−0.31**	−0.06	0.06
Resid	−0.23	0.24	−0.08
*R*^2^(%)	36.6	15.9	10.9

^†^ Bold fonts are those having eigenvectors > |0.25|. ^‡^ Abbreviation for the traits, see Table 2 and Appendix A.

## Data Availability

The data presented in this study are available on request from the corresponding author.

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
