# Peer review of "High Resistant Starch Rice: Variation in Starch Related SNPs, and Functional, and Sensory Properties"

_foods, 2021, doi:10.3390/foods11010094_

Round 1

Reviewer 1 Report

keywords: to the word "functional" she added was the "property"

A very well written introduction. Reference to the importance of resistant starch in terms of nutrition and digestive processes, which is due to the properties and reactivity of the starch.

The continuations of tables 2 and 4 used in the work should be separated as successive, new tables. 

line 212, 227 - where is Table S2. 

For confirmation of the reliability of the tests in chapter 3.3. Association of grain quality traits with RVA parameters - selected (the most interesting, extremely different) - viscosity curves in the temperature and time gradient should be included, the authors of which write in the methodology 

line 170- The rice samples were cooked in a rice cooker (Aroma ARC-787D-1NG, San Diego, 170 CA) until automatically stopped, indicating all water had been absorbed. - necessary detailing of cooking parameters, time, temperature, etc. 

2.2. Apparent amylose content - the assumption of the methodology description is such that the potential reader can repeat the experience according to the guidelines contained in the articles. Therefore, it suggests complementing the method of quantifying amylose.

2.3. Protein - it is worthwhile to degrease rice flour for protein determination, specify the fat content. As the Soxtext method is a standard method for fat determination - one more detail of the method would be necessary.

Why was the generally recognized and accepted kiejdahl method not used for protein determination? 

Reviewer 2 Report

This manuscript is interesting and has potential utility. There are few comments to be addressed:

Abstract part needs to be rewritten; in the abstract part, add an objective, main results, general conclusion, and more numerical data. For the first time, you cannot use uncommon abbreviations.

First describe full form before using abbreviations. 

Would be interesting to draw a Spider web graph/diagram of texture attributes.

Lines 422-423: "Therefore, high RS rice might find more acceptance by consumers in certain global regions compared to others." - please exemplify

Conclusion: avoid small paragraphs, please merge them and interrelate them for good reading.

Reviewer 3 Report

This is an interesting and valuable manuscript describing the effects of genetics and resistant starch on the functional and cooked rice texture properties for purpose of breeding for higher resistant starch rice of specific quality. But the following points need to be done by the authors:

  • The abstract should be more informative by giving real results rather than elastic sentences. Important and main contents should be given. Support the results with some quantitative data. Moreover, no conclusions are provided.
  • Introduction: It was concluded that the diverse functional ……………. property and cooked rice texture. => The conclusion is not mentioned in the introduction of the article. This section should be presented in the conclusion section. Please remove it.
  • The purpose of the research should be stated more clearly at the end of the introduction.
  • Line 126: All physicochemical and sensory analyses for each sample were determined in duplicate. => How to determine the standard deviation and significant letters with two repetitions?
  • Result and Discussion: This part needs more specific detailed comparative studies. Please compare with similar works after presenting each result. Please correct in the whole text.
  • Bivariate correlation coefficients of grain quality traits and texture attributes were not high value. These small values cannot be considered as a correlation. Please correct the whole of the manuscript.

Round 2

Reviewer 3 Report

Thanks for corrections and your reply.